# Symptom reporting, healthcare-seeking behaviour and antibiotic use for common infections: protocol for Bug Watch, a prospective community cohort study

Catherine M Smith,[1] Anne Conolly,[2] Christopher Fuller,[1] Suzanne Hill,[2] Fabiana Lorencatto,[3] Franziska Marcheselli,[2] Susan Michie,[3] Jennifer S Mindell,[4] Matthew J Ridd,[5] Laura J Shallcross,[1] Georgios Tsakos,[6] Andrew Hayward,[6] Ellen B Fragaszy[1,7]

For numbered affiliations see end of article.

**Correspondence to**
Dr Catherine M Smith;
catherine.m.smith@ucl.ac.uk

## ABSTRACT

**Introduction** Antimicrobial resistance is a significant worldwide problem largely driven by selective pressure exerted through antibiotic use. Preserving antibiotics requires identification of opportunities to safely reduce prescriptions, for example in the management of mild common infections in the community. However, more information is needed on how infections are usually managed and what proportion lead to consultation and antibiotic use. The aim of this study is to quantify consultation and prescribing patterns in the community for a range of common acute infection syndromes (respiratory, gastrointestinal, skin/soft tissue, mouth/dental, eye and urinary tract). This will inform development of interventions to improve antibiotic stewardship as part of a larger programme of work, Preserving Antibiotics through Safe Stewardship.

**Methods and analysis** This will be an online prospective community cohort study in England. We will invite 19 510 adults who previously took part in a nationally representative survey (the Health Survey for England) and consented to be contacted about future studies. Adults will also be asked to register their children. Data collection will consist of a baseline registration survey followed by weekly surveys sent by email for 6 months. Weekly surveys will collect information on symptoms of common infections, healthcare-seeking behaviour and use of treatments including antibiotics. We will calculate the proportions of infection syndromes that lead to General Practitioner consultation and antibiotic prescription. We will investigate how healthcare-seeking and treatment behaviours vary by demographics, social deprivation, infection profiles and knowledge and attitudes towards antibiotics, and will apply behavioural theory to investigate barriers and enablers to these behaviours.

**Ethics and dissemination** This study has been given ethical approval by the University College London Research Ethics Committee (ID 11813/001). Each participant will provide informed consent upon registration. We will disseminate our work through publication in peer-reviewed academic journals. Anonymised data will be

### Strengths and limitations of this study

► This study will use a novel and efficient method for large-scale collection of information about symptoms and related healthcare-seeking behaviours.
► It will address an important aspect of primary care (antibiotic use) by collecting data on a comprehensive set of symptoms of common infections, but will not cover sexually transmitted infections.
► Collecting data over an entire year will allow seasonal variations to be explored.
► Participants will be recruited from a sample that is representative of the population living in private households in England.
► The prospective community cohort design will enable information to be captured about symptoms irrespective of medical consultation.

made available through the UK Data Service (https://www.ukdataservice.ac.uk/).

## INTRODUCTION

The emergence of antimicrobial resistance (AMR) is a significant worldwide problem. It is largely driven by selective pressure exerted through antibiotic use and has led to some infections becoming untreatable with existing antimicrobials.[1] Preserving antibiotics for the future depends on achieving a safe balance between potential population harms of prescribing antibiotics and risks to the individual of not prescribing. The set of actions that promote this responsible use of antibiotics is referred to as antibiotic stewardship.[2]

Improving stewardship requires identification of opportunities to safely reduce prescriptions of antibiotics. Although

overprescribing of antibiotics for patients presenting at primary care with common infections has been widely reported,[3] there is also evidence for a significant clinical 'iceberg' of infection.[4 5] For example, previous studies have shown that most patients safely manage respiratory and gastrointestinal symptoms without consulting their General Practitioner (GP) or taking an antibiotic.[4 5] This suggests that inappropriate antibiotic prescriptions could be reduced through improved management of common infections and associated symptoms in the community. There is limited information, however, on how the public manage symptoms of other infections, what proportion of infections lead to consultation and antibiotic use, or how these rates vary according to type of symptoms or patient characteristics. Establishing this requires information to be captured on patients in the community, including those who do not seek healthcare, and identification of healthcare-seeking behaviours, ideally through prospective follow-up.

The aim of the Bug Watch study is to quantify consultation and antibiotic prescribing patterns in the community for a range of acute common infections (respiratory, gastrointestinal, skin/soft tissue, mouth/dental, eye and urinary tract). Bug Watch is part of a larger programme of work, Preserving Antibiotics through Safe Stewardship (PASS), which aims to inform the development of multifaceted behavioural interventions that will strengthen antibiotic stewardship across a range of healthcare settings. Results from Bug Watch will be synthesised with insights from qualitative interviews to identify opportunities for improved antibiotic stewardship in the community and general practice and inform development of interventions.

## METHODS AND ANALYSIS
### Study design and setting
This will be an online prospective community cohort study in England. Data collection will consist of a baseline registration survey followed by weekly surveys for 6 months. Weekly surveys will collect information on symptoms of common infections, healthcare-seeking behaviour and use of treatments including antibiotics.

### Recruitment
We will recruit participants through the Health Survey for England (HSE). HSE is an annual survey first conducted in 1991 that monitors changes in the health and lifestyles of people living in England.[6] It is commissioned by National Health Service (NHS) Digital and run by NatCen Social Research and the University College London (UCL) Research Department of Epidemiology and Public Health. The sample of individuals included in HSE each year is designed to be representative of the population living in private households in England. Full details of the methods for HSE sample have been described previously.[7]

We will invite all adults who took part in the HSE in 2013, 2014 or 2015 and who consented to be contacted about future research studies (with the exception of adults aged 50–53 years from the 2015 survey who were recruited to a different study). This comprises 19 510 adults, of whom we estimate 15 819 (81%) will still be resident at the address on record when they are contacted. Parents or guardians will be invited to register up to four children aged under 16, and all information for children will be reported by the adult who registered them. Anyone aged 16 or over living in the same household will be invited to register separately.

As this is a largely descriptive study, and the primary outcome is calculation of rates, we did not require a formal power calculation. Based on their experience of previous population surveys, NatCen estimated that approximately 25% of those who received a letter would sign up (~4000), and approximately 50% of those who signed up would complete follow-up (~2000 people completing 6 months of follow-up). This would equate to approximately 1000 person-years of follow-up, which would allow calculation of crude rates of 500 cases per 1000 person-years with a 95% CI of 457 to 546; 200 cases per 1000 person-years, 95% CI 173 to 230; and 100 cases per 1000 person-years 95% CI 81 to 122.

Recruitment will be conducted in four waves starting in March, June, September and November 2018. Initial invitations to take part in Bug Watch will be sent by post. Those who wish to sign up will be directed to a web link. A second reminder letter will be sent to those who have not registered approximately 3 weeks after the initial invitation was sent. Where possible, invitation reminders will also be sent by email or SMS text message. Individuals who are invited and do not wish to take part in the study will be removed from contact lists.

### Participant materials and incentives
The initial invitation letter will include a participant information sheet which will describe the purpose of the study, what participants will be asked to do and the contact details of the study team. It will also include a paper copy of a 'symptom diary', which will be designed to help participants to keep track of the symptoms of infection that will be collected in the weekly surveys.

Study invitations will direct those interested in participating to a UCL web page. This page will include the registration survey URL as well as a short video describing the study. Those wishing to register children will also be able to download an information sheet for children. We will design this sheet so that it is suitable to be read by anyone aged approximately 8 years or older. Following registration, participants will be sent a reusable laminated copy of the symptom diary, a pen and a Bug Watch-branded magnet.

During follow-up, participants will be sent two email newsletters. These newsletters will provide updates about the study such as rates of weekly survey completion. To avoid influencing the behaviour of participants during follow-up, the newsletters will not include details about the main study outcomes.

All participants will receive a £5 voucher to thank them for registering. At the end of follow-up, those who have completed at least 50% of their weekly surveys will be sent an additional £5 voucher to thank them for participating. They will also be entered into a prize draw to win a further £50 voucher.

## Data collection

All data will be collected using Research Electronic Data Capture (REDCap) electronic data capture tools hosted on the UCL Data Safe Haven.[8] REDCap is a secure, web-based application for research studies. The UCL Data Safe Haven provides a technical solution for storing, handling and analysing identifiable data. It has been certified to the ISO27001 information security standard and conforms to NHS Digital's Information Governance Toolkit. Data analysis will also be conducted within the UCL Data Safe Haven, from which personal information will not be removed.

Online registration will take approximately 15 min/person. Participants will be asked to provide consent, confirming that they have read and understood the information sheets. Parents or legal guardians will be asked to give formal consent for children under the age of 16. For children that are able to give assent, the parent or legal guardian will be asked to discuss with them whether they wish to take part or not. Consent for linking data to the broader HSE survey data and for contact for qualitative interviews will also be requested.

Full details of the baseline data collection are shown in table 1. The baseline survey will collect information on contact details, demographics, household composition, general health, oral health and GP consultations. It will also include questions about knowledge and attitudes towards antibiotics, adapted from Wave 3 of the Wellcome Trust Monitor survey, and the EQ-5D-3L instrument for measuring health-related quality of life.[9 10] The Wellcome

| Table 1 | Baseline data collection |
|---|---|
| Section | Fields included |
| Consent | ▶ Consent to participate in Bug Watch (required).<br>▶ Permission to be contacted for qualitative interviews; for data to be linked to the Health Survey for England; to be contacted about a urinary tract infection substudy (optional). |
| Contact details | ▶ ID number (from invitation letter).<br>▶ Name.<br>▶ Email address.<br>▶ Postal address. |
| Demographics | ▶ Date of birth.<br>▶ Sex.<br>▶ Country of birth.<br>▶ Ethnic group.<br>▶ Work status (employed, in education, unemployed, retired and so on).<br>▶ Full or part time work.<br>▶ Is a healthcare worker. |
| General health | ▶ Long term illnesses or health problems.<br>▶ Recurrent urinary tract infections.<br>▶ Currently pregnant; which trimester.<br>▶ Smoking status.<br>▶ Seasonal influenza vaccine in the last year.<br>▶ EQ-5D-3L. |
| GP consultations | ▶ Number of GP consultations in last 12 months. |
| Antibiotics | ▶ Ever been prescribed antibiotics; number in last 12 months.<br>▶ Ever been prescribed antibiotics but thought it was not the right treatment.<br>▶ When last took antibiotics; were they prescribed (if not, where from); were all taken.<br>▶ Ever asked for an antibiotic prescription; was it given; needed to persuade.<br>▶ Which conditions think can be treated with antibiotics.<br>▶ Understanding of term 'antibiotic resistance'. |
| Oral health | ▶ Rate dental health (global item).<br>▶ Has dentures.<br>▶ Dental symptoms in last 12 months.<br>▶ Problems caused by mouth/teeth/dentures in last 12 months (impact on quality of life). |
| Household composition | ▶ Number of adults (aged 16+).<br>▶ Number of children, number to be registered (up to 4). |

NB, Questions are filtered and adapted based on previous responses so that they are only shown to participants when relevant. For example, 'Currently pregnant' is not shown if sex is given as male. GP, General Practitioner.

Trust Monitor survey is designed to be representative of the UK adult population and measures trends in public attitudes towards science. Wave 3 of the survey, conducted in 2015, included a set of questions about knowledge and attitudes towards antibiotics. The EQ-5D-3L is a standardised measure of health status developed by the EuroQol Group that provides a simple, generic measure of health for clinical and economic appraisal.[10]

On the first Monday after registration, participants will be sent an email reminding them to start keeping track of symptoms of infection. Following this, they will be sent weekly emails each Monday with a link to a survey to fill in details about symptoms from the previous 7 days and how they managed them. These emails will be sent each week for 6 months, and each survey will be open for 1 week. One reminder email will be sent on the Thursday of each week if the survey has not been completed. Participants will be given the opportunity to complete feedback forms about Bug Watch after approximately 6 weeks of participation, and again after all weekly surveys have been sent.

The symptoms that will be monitored in Bug Watch are shown in table 2. For each day that a symptom is reported, participants will be asked to rate its severity (mild, moderate or severe) and to complete the EQ-5D-3L. They will also be asked to report what they did about their symptoms including whether they took time off work (or school), sought medical advice, or took any treatments. When antibiotics are reported, further details will be collected about the type of antibiotic, duration, who it was prescribed by (or how otherwise obtained) and adherence to treatment.

At the end of a series of symptoms, participants will be asked to complete a set of questions about what influenced how they managed their symptoms, specifically whether or not they consulted their GP and sought antibiotics. We will apply the Capability, Motivation, Opportunity, Behaviour (COM-B) model of behaviour change, which has been validated for studies of the general population,[11] to explore the wide-range of potential individual, sociocultural and environmental barriers/enablers to these behaviours. The survey will include at least one item mapping onto each domain of the COM-B model (table 3). Items will be in the form of belief statements to which participants rate their agreement on Likert-type scales from 1—Strongly Disagree to 5—Strongly Agree.

In all surveys, we will make use of 'skip logic' to ensure that participants are only shown questions that are relevant to them. For example, they will first be asked if they have any symptoms, and only if they do will they be shown specific symptom categories. Weekly surveys will therefore take no longer than a minute if no symptoms are reported, and approximately 5–10 min if symptoms are reported.

## Statistical analysis

We will describe the demographic characteristics of the Bug Watch cohort and assess its representativeness by comparing with the characteristics of the broader HSE

**Table 2** Symptoms of infection to be collected in Bug Watch

| Respiratory | Gastrointestinal | Eye | Urinary tract | Skin/soft tissue | Mouth/dental | General/non-specific |
|---|---|---|---|---|---|---|
| Runny nose | Nausea | Red eye | Painful urination | Rash (general) | Toothache | Fever |
| Blocked nose | Vomiting | Conjunctivitis | Frequent urination | Rash (local) | Mouth ulcer | Chills |
| Sneezing | Stomach/abdominal | Stye | Urgent urination | Itchy | Gum abscess | Muscle aches |
| Dry cough | pain | | Cloudy/dark/smelly | (general) | | Night sweats |
| Coughing up phlegm | Diarrhoea | | urine | Itchy (local) | | Fatigue |
| Short of breath | | | Blood in urine | Boils/abscesses | | Headache |
| Ear ache/pain | | | Bladder pain | Infected wound/cut | | Migraine |
| Fluid leaking from ear | | | Kidney pain | Mastitis | | Loss of appetite |
| Sinus Pain/congestion | | | | Chicken pox | | |
| | | | | Shingles | | |

**Table 3** Example items exploring barriers/enablers to GP consulting and antibiotic seeking behaviours, based on the COM-B model of behaviour change[11] (asked at the end of a series of symptoms in Bug Watch)

| COM-B domain | Example barrier/enabler belief statements |
|---|---|
| Capability (psychological) | 'I thought antibiotics would be effective in treating my symptoms'<br>'I did not know what other treatments were available' |
| Capability (physical) | 'I felt too unwell to travel to the GP practice' |
| Opportunity (social) | 'I was encouraged by others to go see my GP'<br>'My GP discussed alternatives ways of managing my symptoms'<br>'I was involved in the decision of whether or not to take antibiotics' |
| Opportunity (physical) | 'I was unable to take time off work to recover without taking antibiotics'<br>'Other treatments were too expensive'<br>'It was easy to get a GP appointment' |
| Motivation (reflective) | 'I felt confident in safely treating my symptoms without antibiotics'<br>'I did not think I would get better as quickly without antibiotics' |
| Motivation (automatic) | 'I was worried about my symptoms'<br>'I always go see my GP when I have these types of symptoms'<br>'I felt reassured that I could safely manage my symptoms without antibiotics' |

COM-B, Capability, Motivation, Opportunity, Behaviour; GP, General Practitioner.

sample and published national statistics. We will also assess the representativeness of the knowledge and attitudes towards antibiotics of the cohort by comparing with the Wellcome Trust Monitor Survey.[9]

Symptoms reported will be combined into infection syndromes (ie, combinations of symptoms associated with an episode of infection). We will create descriptive profiles of infection syndromes including types of symptom reported, duration, timing and severity. We will calculate the proportions of infection syndromes that lead to people consulting their GP and receiving antibiotics. Antibiotic use will be described in terms of type of antibiotic, duration, who it was prescribed by (or how otherwise obtained) and adherence to treatment. Other healthcare-seeking behaviours, behavioural influences (barriers and enablers within the COM-B model) and treatments taken to manage symptoms will also be described.

We will use Poisson regression to calculate rates of infection, consultation and antibiotic prescribing. These analyses will be weighted by the population structure of England (age, sex, index of multiple deprivation, region—as indicted by representativeness) and will account for the clustered nature of the data. We will use logistic regression to investigate how GP consultation and antibiotic prescribing varies by age, gender, ethnicity, presence of other illnesses, social deprivation, infection syndrome and knowledge and attitudes towards antibiotics. Continuous variables will be converted into categorical variables. We will also assess the impact of different types of infection on quality of life using the EQ5D-3L scores, and on reported work and school absences. We will use these measures to estimate the overall impact of community infection at the population level. Full statistical methods will be presented with relevant analyses.

### Patient and public involvement

Participants were not directly involved in design of this study although feedback will be collected at two time points during follow-up. Newsletters will be sent to give participants updates during the study, and a summary of the main findings will be available at the end. As part of the wider PASS study, a subset of participants will be invited to take part in qualitative interviews that will draw on behavioural theory to investigate the drivers of healthcare-seeking behaviours (full methods will be published elsewhere). Findings from Bug Watch and the related interviews will inform stakeholder panels to develop stewardship interventions through a user-centred design approach.

### Ethics and dissemination

Each participant will provide informed consent on registration. We will disseminate our work through publication in peer-reviewed academic journals and presentation at conferences. Findings will contribute to interventions and educational materials developed through the wider PASS study. Anonymised data from Bug Watch will be made available through the UK Data Service (https://www.ukdataservice.ac.uk/) for use by other researchers.

**Author affiliations**
[1]Institute of Health Informatics, University College London, London, UK
[2]NatCen Social Research, London, UK
[3]Centre for Behaviour Change, University College London, London, UK
[4]Department of Epidemiology and Public Health, University College London, London, UK
[5]Population Health Science Institute, University of Bristol, Bristol, UK
[6]Institute of Epidemiology and Health Care, University College London, London, UK
[7]Faculty of Epidemiology & Population Health, London School of Hygiene and Tropical Medicine, London, UK

**Contributors** Written on behalf of the Preserving Antibiotics through Safe Stewardship (PASS) study research group. CMS is responsible for data collection, management and analysis, and drafted the protocol. AH, EBF and the PASS study

investigators developed the concept for the study. All authors (CMS, AC, CF, SH, FL, FM, SM, JSM, MJR, LS, GT, AH, EBF) contributed to the study design and development of the questionnaires. AC, FM and SH led recruitment. All authors (CMS, AC, CF, SH, FL, FM, SM, JSM, MJR, LS, GT, AH, EBF) have checked and approved the protocol to be published.

**Funding** This work is supported by the Economic and Social Research Council, grant number ES/P008321/1, as part of the Preserving Antibiotics through Safe Stewardship (PASS) project. Professor Hayward is a National Institute for Health Research (NIHR) Senior Investigator. MJR is funded by an NIHR Post-Doctoral Research Fellowship (PDF-2014-07-013). The views expressed in this publication are those of the authors and not necessarily those of the NHS, the National Institute for Health Research or the Department of Health and Social Care.

**Competing interests** None declared.

**Patient consent for publication** Not required.

**Ethics approval** This study has been given ethical approval by the UCL Research Ethics Committee (ID 11813/001).

**Provenance and peer review** Not commissioned; externally peer reviewed.

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
