## [Reviewer comments · BMJ Open]

ARTICLE DETAILS

TITLE (PROVISIONAL)	Symptom reporting, healthcare-seeking behaviour and antibiotic use for common infections: protocol for Bug Watch, a prospective community cohort study
AUTHORS	Smith, Catherine; Conolly, Anne; Fuller, Christopher; Hill, Suzanne; Lorenatto, F; Marcheselli, Franziska; Michie, Susan; Mindell, Jennifer; Ridd, Matthew; Shallcross, Laura; Tsakos, Georgios; Hayward, Andrew; Fragaszy, Ellen

VERSION 1 - REVIEW

REVIEWER	Lin Yang The Hong Kong Polytechnic University
REVIEW RETURNED	28-Jan-2019

GENERAL COMMENTS	The authors plan to conduct a community cohort study by inviting the participants to conduct monthly online surveys on infectious diseases and healthcare seeking behavior. This is an interesting study to address a very important topic. I have some suggestions for their consideration. 1. The major concern is that they got an ambitious plan to cover a variety of nonspecific symptoms for a wide range of acute infections, making it difficult to link to specific bacterial or viral infections and judge whether antibiotics usage is appropriate. For example, adenovirus may cause upper respiratory symptoms and red eyes, and many respiratory pathogens cause both respiratory and gastrointestinal symptoms.2. Another challenge lies in knowledge levels of participants. Most people may not be able to differentiate antimicrobial from other medications. The authors did not provide details on how they are dealing with this problem. For, it is not surprising for nonprofessionals to mix the allergy with infections especially for skin or respiratory symptoms. At least the authors could collect their allergy and medication history in the baseline survey.3. The follow-up period for each participant is 6 months, I understand this could reduce report fatigue but I am concerned the period is too short to address the seasonal variations. Since the incentives are provided at the end of study, can you consider extending the individual follow-up to one year instead? As the study period is one year, what about the participants recruited at the fourth quarter? Will they also be followed for six months?4. More details about the focus group interview should be provided in this protocol. How many participants do you plan to recruit for
--

	this interview? What is behavioral theory? What are the potential drivers proposed in previous studies? 5. It is more appropriate to use antimicrobial than antibiotics in this paper since the authors cover both bacterial, fungal and viral infections. The term "health-seeking behaviours" should be "healthcare-seeking behavior" throughout the text.
--	--

REVIEWER	Minyon Avent Infection and Immunity Theme UQ Centre for Clinical Research The University of Queensland Australia
REVIEW RETURNED	01-Feb-2019

GENERAL COMMENTS	Dear Editor, Thank you for asking me to review the study entitled 'Symptom reporting, healthcare-seeking behaviour and antibiotic use for common infections: protocol for Bug Watch, a prospective community cohort study '. The proposal has been well written and clearly describes the aim of the study.
---

REVIEWER	Koen Pouwels Public Health England
REVIEW RETURNED	13-Feb-2019

GENERAL COMMENTS	The manuscript describes a protocol for an interesting observational study. However, the current evidence should be more clearly described and referenced and more details about the statistical methodology are needed. Introduction: The introduction should explain the rationale for the study and what evidence gap it may fill. Currently there is not enough information about what is already known from other studies. For example, the likelihood of consulting for and asking for antibiotics related to RTI has been investigated by several other groups: https://www.ncbi.nlm.nih.gov/pubmed/23834879 & https://www.eurosurveillance.org/content/10.2807/1560-7917.ES.2018.23.25.1700424 The proportion of consultations that result in an antibiotic prescription has been investigated for a much wider range of common infections in a large retrospective study from England: https://academic.oup.com/jac/article/73/suppl_2/19/4841820 How these proportions vary over patient characteristics, e.g. comorbidities and gender has also been assessed previously: https://academic.oup.com/jac/article/73/suppl_2/19/4841820 & https://bmjopen.bmj.com/content/8/2/e020203?cpetoc=&utm_source=TrendMD&utm_medium=cpc&utm_campaign=BMJOp_TrendMD-0 The consultation rate for RTIs has also been shown to decline in recent years: https://www.ncbi.nlm.nih.gov/pubmed/29955785 Methods:
--

	Publishing protocols makes it easier to spot and understand any deviations from protocol / data fishing exercises. To be useful a protocol should therefore include detailed information about the methodological and statistical approaches that are going to be used. The current protocol is not clear enough about what methods are going to be used. For example, the authors only report that they will assess representativeness of the Bug Watch cohort by comparing with other sources. What is lacking is how they are going to compare this (e.g. standardized differences or some other measure?) and what they will do if it appears not to be representative (e.g. are the authors attempting to estimate population average treatment effects (PATE) and if so, how are they going to construct and apply the weights). The sentence 'We will use these measures to estimate the overall impact of community infection at the population level' suggest that the authors aim the estimate PATE, so more detail is needed how weights will be constructed and applied. Similarly, it is not sufficient to just state that regression methods will be used, this leaves to many degrees of freedom for the researchers to try various models and select the one with the most appealing results. What type of regression will be used for each outcome & what are the specific outcomes, how is each covariate handled and what does it mean (e.g. what is infection profile), how will potential non-linear relationships of continuous covariates with the outcome be handled, on what basis will the authors decide which variables should be included in each specific regression model, will interactions between variable assessed and if so which on what basis, etc. I couldn't find any sample size calculation. As such it is unclear to what extent the authors can actually answer the questions they like to answer. Given that there are weekly surveys to complete, I can imagine that there will be significant dropout. Did the authors pilot this or have previous experience that could indicate whether this may not be a problem. Personally, another 5 pound at the end of the study wouldn't be a convincing argument for me to regularly complete the weekly surveys.
--	---

REVIEWER	Peter Herbison University of Otago New Zealand
REVIEW RETURNED	15-Feb-2019

GENERAL COMMENTS	I have only a couple of comments. The first is that some of the things the authors say they will report on require the sample to be representative of the population. This is a self selected sample from a representative sample, that is supplemented by children. So how are they going to assure people that those things that rely on a representative sample are close to the truth? The protocol implies that the EQ-5D-3L will be filled in for every symptom that the participants have, even if multiple symptoms are entered at the same time. As the EQ-5D-3L is a generic instrument (as they say) it is not specific to a particular symptom and so should be the same for multiple simultaneous symptoms.
--

REVIEWER	Yanhong Jessika hu The University of Hong Kong
REVIEW RETURNED	16-Feb-2019

GENERAL COMMENTS	This is a very important study to look at the population level to define the antibiotic use behaviour factors by using COMB theory related to symptoms, health-seeking behaviour, social and geographic factors. Here are the key concerns: 1) Feasibility: although the existing established surveillance system will be helpful, there may be problems with the use of this system, such as... a) reporting bias: remembering symptoms and antibiotic use in the last 12 months is difficult. I understand many published papers used the time period, however, it seems to me too long for people to remember. On the other hand, people may not have any symptoms in the past 1 month nor 3 months. Is there preliminary data to demonstrate the rationale for using a 12 month reporting window? Otherwise, if there is a written record which they can refer to, that might be helpful. Since the study will follow up for 6 months, the past 12 months might not be the best choice to avoid the reporting bias. b) sample size: there are many outcomes and measurements - which one is the primary outcome? Authors did not state the power of recruiting 19,510 for 6 major symptoms, it will be useful for readers to know if this sample is sufficient to reach the desired power. c) recruitment: what rate is expected from the study? Will the 5 pounds incentive and the lottery draw of 50 pounds incentives provide sufficient motivation? A flow chart might be better to make the recruitment process more clear. 2) COMB theory: Has this strategy been validated for this problem in this population?
---

VERSION 1 – AUTHOR RESPONSE

Reviewer 1

The authors plan to conduct a community cohort study by inviting the participants to conduct monthly online surveys on infectious diseases and healthcare seeking behavior. This is an interesting study to address a very important topic. I have some suggestions for their consideration.

1. The major concern is that they got an ambitious plan to cover a variety of nonspecific symptoms for a wide range of acute infections, making it difficult to link to specific bacterial or viral infections and judge whether antibiotics usage is appropriate. For example, adenovirus may cause upper respiratory symptoms and red eyes, and many respiratory pathogens cause both respiratory and gastrointestinal symptoms.

We agree that the symptoms reported in this study may be indicative of bacterial or viral infections in many instances. However, in this study, our aim is to quantify consultation and prescribing patterns for common infections based on reported symptoms, rather than to determine whether prescribing is appropriate. The symptoms reported reflect the information that is available to the patient when deciding whether to consult; and to the GP when deciding whether to prescribe antibiotics (as laboratory testing is used relatively infrequently in the community setting).

2. Another challenge lies in knowledge levels of participants. Most people may not be able to differentiate antimicrobial from other medications. The authors did not provide details on how they are dealing with this problem. For, it is not surprising for nonprofessionals to mix the allergy with infections especially for skin or respiratory symptoms. At least the authors could collect their allergy and medication history in the baseline survey.

Although we acknowledge that there will be times when a participant is unsure if the treatment that they are taking is an antibiotic, pilot data has indicated that this is not a serious issue in this study. Since the surveys are completed online, participants are able to check the type of medication that they have if necessary. Our baseline data collection includes reporting of chronic conditions.

3. The follow-up period for each participant is 6 months, I understand this could reduce report fatigue but I am concerned the period is too short to address the seasonal variations. Since the incentives are provided at the end of study, can you consider extending the individual follow-up to one year instead? As the study period is one year, what about the participants recruited at the fourth quarter? Will they also be followed for six months?

We made a pragmatic decision to follow participants up for six months, prioritising retention of participation over length of follow-up, and allowing the study to be completed during its funding cycle. We have also staggered recruitment in four waves to ensure good coverage over the calendar year and allow us to assess seasonal variations. All participants will be followed-up for six months.

4. More details about the focus group interview should be provided in this protocol. How many participants do you plan to recruit for this interview? What is behavioral theory? What are the potential drivers proposed in previous studies?

The qualitative interviews will be conducted as part of the wider programme of work (Preserving Antibiotics through Safe Stewardship), but full protocol and methods will be presented elsewhere. This has been clarified in the Patient and public involvement section (p6):

“As part of the wider PASS study, a subset of participants will be invited to take part in qualitative interviews that will draw on behavioural theory to investigate the drivers of healthcare-seeking behaviours (full methods will be published elsewhere).”

5. It is more appropriate to use antimicrobial than antibiotics in this paper since the authors cover both bacterial, fungal and viral infections. The term “health-seeking behaviours” should be “healthcare-seeking behavior” throughout the text.

We have used the terminology “antibiotics” to reflect the phrasing used in the survey – which asks participants specifically about antibiotics, rather than antimicrobials more broadly. We have corrected the instances where the text read “health-seeking” to “healthcare-seeking”.

Reviewer 2

Thank you for asking me to review the study entitled 'Symptom reporting, healthcare-seeking behaviour and antibiotic use for common infections: protocol for Bug Watch, a prospective community cohort study '. The proposal has been well written and clearly describes the aim of the study.

Reviewer 3

The manuscript describes a protocol for an interesting observational study. However, the current evidence should be more clearly described and referenced and more details about the statistical methodology are needed.

Introduction:

1. The introduction should explain the rationale for the study and what evidence gap it may fill. Currently there is not enough information about what is already known from other studies. For example, the likelihood of consulting for and asking for antibiotics related to RTI has been investigated by several other groups:

<https://www.ncbi.nlm.nih.gov/pubmed/23834879>

&

<https://www.eurosurveillance.org/content/10.2807/1560-7917.ES.2018.23.25.1700424>

The proportion of consultations that result in an antibiotic prescription has been investigated for a much wider range of common infections in a large retrospective study from England

https://academic.oup.com/jac/article/73/suppl_2/19/4841820

How these proportions vary over patient characteristics, e.g. comorbidities and gender has also been assessed previously:

https://academic.oup.com/jac/article/73/suppl_2/19/4841820

&

https://bmjopen.bmj.com/content/8/2/e020203?cpetoc=&utm_source=TrendMD&utm_medium=cpc&utm_campaign=BMJOp_TrendMD-0

The consultation rate for RTIs has also been shown to decline in recent years:

<https://www.ncbi.nlm.nih.gov/pubmed/29955785>

Whilst previous studies have calculated prescription rates in patients who consult primary care with suspected clinical infection syndromes, Bug Watch recruits individuals in the community and therefore includes individuals with symptoms of infection who do not consult their GP. This will enable calculation of rates of consultation and prescribing among patients with symptoms of infection in the community, and importantly provides insight into the types of patient (and symptom profiles) of patients who do and do not seek healthcare. Two previous prospective studies - Flu Watch (Hayward, 2014) and IID2 (Tam, 2012) – have taken a similar approach, looking specifically at respiratory and gastrointestinal infections respectively. However, there has been no systematic approach to estimate these measures across a range of acute infections. The Bug Watch study will address this gap. We have updated paragraph 2 of our introduction to clarify this:

“Improving stewardship requires identification of opportunities to safely reduce prescriptions of antibiotics. Although overprescribing of antibiotics for patients presenting at primary care with common infections has been widely reported,[3] there is also evidence for a significant clinical “iceberg” of infection.[4,5] For example, previous studies have shown that most patients safely manage respiratory and gastrointestinal symptoms without consulting their GP or taking an antibiotic.[4,5] This suggests that inappropriate antibiotic prescriptions could be reduced through improved management of common infections and associated symptoms in the community. There is

limited information, however, on how the public manage symptoms of other infections, what proportion of infections lead to consultation and antibiotic use, or how these rates vary according to type of symptoms or patient characteristics. Establishing this requires information to be captured on patients in the community, including those who do not seek healthcare, and identification of healthcare-seeking behaviours, ideally through prospective follow-up.“

Methods:

2. Publishing protocols makes it easier to spot and understand any deviations from protocol / data fishing exercises. To be useful a protocol should therefore include detailed information about the methodological and statistical approaches that are going to be used. The current protocol is not clear enough about what methods are going to be used. For example, the authors only report that they will assess representativeness of the Bug Watch cohort by comparing with other sources. What is lacking is how they are going to compare this (e.g. standardized differences or some other measure?) and what they will do if it appears not to be representative (e.g. are the authors attempting to estimate population average treatment effects (PATE) and if so, how are they going to construct and apply the weights). The sentence ‘We will use these measures to estimate the overall impact of community infection at the population level’ suggest that the authors aim the estimate PATE, so more detail is needed how weights will be constructed and applied.

Similarly, it is not sufficient to just state that regression methods will be used, this leaves to many degrees of freedom for the researchers to try various models and select the one with the most appealing results. What type of regression will be used for each outcome & what are the specific outcomes, how is each covariate handled and what does it mean (e.g. what is infection profile), how will potential non-linear relationships of continuous covariates with the outcome be handled, on what basis will the authors decide which variables should be included in each specific regression model, will interactions between variable assessed and if so which on what basis, etc.

We will assess the representativeness of the Bug Watch cohort by comparing baseline data with published national statistics and the wider Health Survey for England sample (demographics); and from the Wellcome Trust Monitor Survey (antibiotic knowledge and attitudes) (see first paragraph under Statistical analysis). For analyses of rates, we will use Poisson regression weighted by the population structure of England (age, sex, index of multiple deprivation, region – as indicted by representativeness). For analyses of variation in consultation and antibiotic prescribing and assessment of risk factors, we will use unweighted logistic regression. We have updated paragraph 3 of under Statistical analysis to include these details:

“We will use Poisson regression to calculate rates of infection, consultation and antibiotic prescribing. These analyses will be weighted by the population structure of England, and will account for the clustered nature of the data. We will use logistic regression methods to investigate how healthcare-seeking GP consultation and antibiotic prescribing and treatment behaviours vary varies by age, gender, ethnicity, presence of other illnesses, social deprivation, infection syndrome, and knowledge and attitudes towards antibiotics. Continuous variables will be converted into categorical variables. We will also assess the impact of different types of infection on quality of life using the EQ5D-3L scores, and on reported work and school absences, and use these measures to estimate the overall impact of community infection at the population level. Full statistical methods will be presented with relevant analyses.”

3. I couldn't find any sample size calculation. As such it is unclear to what extent the authors can actually answer the questions they like to answer. Given that there are weekly surveys to complete, I can imagine that there will be significant dropout. Did the authors pilot this or have previous experience that could indicate whether this may not be a problem. Personally, another 5 pound at the end of the study wouldn't be a convincing argument for me to regularly complete the weekly surveys.

This is a largely descriptive study, and the primary outcome is to calculate rates rather than measures of association, which is why we did not include a formal power calculation. We discussed expected level of recruitment with collaborators at NatCen (who have considerable experience of population surveys). Approximately 25% of those who received a letter were expected to sign up (~4,000); and approximately 50% of those who signed up were expected to complete follow up (~2,000 people completing six months of follow up). This would equate to ~1,000 person-years of follow up, which would allow calculation of crude rates of 500 cases per 1,000 person-years with a 95% CI of 457-546; 200 cases per 1,000 person-years, 95% CI 173-230; and 100 cases per 1,000 person-years 95% CI 81-122.

The level of incentives discussed used in the study was also discussed with NatCen. The weekly survey submission design was modelled on the Flu Watch study, which had a relatively low drop-out rate (87% reports completed).

Reviewer 4

The first is that some of the things the authors say they will report on require the sample to be representative of the population. This is a self selected sample from a representative sample, that is supplemented by children. So how are they going to assure people that those things that rely on a representative sample are close to the truth?

Please see above response to reviewer 2 (point 3).

The protocol implies that the EQ-5D-3L will be filled in for every symptom that the participants have, even if multiple symptoms are entered at the same time. As the EQ-5D-3L is a generic instrument (as they say) it is not specific to a particular symptom and so should be the same for multiple simultaneous symptoms.

Participants will be asked to fill out the EQ5D-3L for every day that they report symptoms (regardless of how many symptoms are reported), rather than separately for each symptom.

Reviewer 5

This is a very important study to look at the population level to define the antibiotic use behaviour factors by using COMB theory related to symptoms, health-seeking behaviour, social and geographic factors. Here are the key concerns:

1) Feasibility:

although the existing established surveillance system will be helpful, there may be problems with the use of this system, such as...

a) reporting bias: remembering symptoms and antibiotic use in the last 12 months is difficult. I understand many published papers used the time period, however, it seems to me too long for people to remember. On the other hand, people may not have any symptoms in the past 1 month nor 3 months. Is there preliminary data to demonstrate the rationale for using a 12 month reporting window? Otherwise, if there is a written record which they can refer to, that might be helpful. Since the study will follow up for 6 months, the past 12 months might not be the best choice to avoid the reporting bias.

The questions about antibiotic use in the last 12 months are collected at baseline to enable comparisons with the Wellcome Trust Monitor survey, and therefore to assess representativeness of

the knowledge and attitudes towards antibiotics of our sample. Our outcomes will be measured based on the prospective reporting of symptoms, GP consultations, and antibiotic use over the follow-up period.

b) sample size: there are many outcomes and measurements - which one is the primary outcome? Authors did not state the power of recruiting 19,510 for 6 major symptoms, it will be useful for readers to know if this sample is sufficient to reach the desired power.

Please see above response to reviewer 2 (point 2).

c) recruitment: what rate is expected from the study? Will the 5 pounds incentive and the lottery draw of 50 pounds incentives provide sufficient motivation? A flow chart might be better to make the recruitment process more clear.

Please see above response to reviewer 2 (point 3).

2) COMB theory: Has this strategy been validated for this problem in this population?

Bug Watch is a study of the general population of England, and the behaviours investigated are GP consulting and antibiotic seeking. COM-B is a suitable framework to assess this, as it is a generic model of behaviour change and has been validated for studies of the general population (Michie, 2011).

VERSION 2 – REVIEW

REVIEWER	Lin Yang The Hong Kong Polytechnic University
REVIEW RETURNED	16-Mar-2019

GENERAL COMMENTS	The reviewer completed the checklist but made no further comments.
--

REVIEWER	Koen Pouwels University of Oxford, United Kingdom
REVIEW RETURNED	06-Mar-2019

GENERAL COMMENTS	The authors have addressed most of the comments of the reviewers. I've one remark left. Although the authors explain which variables are being used to weight their Poisson regression in their response to the reviewers, they did not incorporate this into the manuscript. It would be much better if this information would also be incorporated into the manuscript.
--

REVIEWER	Peter Herbison University of Otago New Zealand
REVIEW RETURNED	11-Mar-2019

GENERAL COMMENTS	The reviewer completed the checklist but made no further comments.
--

REVIEWER	Yanhong Jessika hu The University of Hong Kong
REVIEW RETURNED	12-Mar-2019

GENERAL COMMENTS	The author addressed the questions that reviewers commented, I would suggest moving those explanations into the main text especially the methods (sample size, power calculation, COMB theory validation) part.
---

VERSION 2 – AUTHOR RESPONSE

Reviewer: 1

I have no further comments.

Reviewer: 3

The authors have addressed most of the comments of the reviewers.

I've one remark left. Although the authors explain which variables are being used to weight their Poisson regression in their response to the reviewers, they did not incorporate this into the manuscript. It would be much better if this information would also be incorporated into the manuscript.

We have now clarified the variables that will be considered for weighting under "Statistical analysis", p6 (new part is highlighted in yellow):

We will use Poisson regression to calculate rates of infection, consultation and antibiotic prescribing. These analyses will be weighted by the population structure of England (age, sex, index of multiple deprivation, region – as indicted by representativeness) and will account for the clustered nature of the data.

Reviewer: 4

I have no further comments on this paper.

Reviewer: 5

The author addressed the questions that reviewers commented, I would suggest moving those explanations into the main text especially the methods (sample size, power calculation, COMB theory validation) part.

We have added the following additional details (highlighted in yellow) to the manuscript:

Sample size/ power calculation: Methods – Recruitment (p4)

As this is a largely descriptive study, and the primary outcome is calculation of rates, we did not require a formal power calculation. Based on their experience of previous population surveys, NatCen estimated that approximately 25% of those who received a letter would sign up (~4,000), and approximately 50% of those who signed up would complete follow-up (~2,000 people completing six months of follow-up). This would equate to approximately 1,000 person-years of follow-up, which would allow calculation of crude rates of 500 cases per 1,000 person-years with a 95% CI of 457-546; 200 cases per 1,000 person-years, 95% CI 173-230; and 100 cases per 1,000 person-years 95% CI 81-122.

COM-B validation: Methods – Data collection (p5)

We will apply the COM-B (Capability, Motivation, Opportunity, Behaviour) model of behaviour change, which has been validated for studies of the general population, [11] to explore the wide-range of potential individual, socio-cultural and environmental barriers/enablers to these behaviours.